# Chemometrics for the Identification of Nitrogen and Acid Compounds in Milk-Whey as By-Products from Crescenza and Grana Padano Type Cheese-Making

**DOI:** 10.3390/molecules26164839

**Published:** 2021-08-10

**Authors:** Stefania Barzaghi, Lucia Monti, Laura Marinoni, Tiziana M. P. Cattaneo

**Affiliations:** 1Research Centre for Animal Production and Aquaculture, Council for Agricultural Research and Economics, Via A. Lombardo, 11, 26900 Lodi, Italy; lucia.monti@crea.gov.it; 2Research Centre for Engineering and Agro-Food Processing, Council for Agricultural Research and Economics, Via G. Venezian, 26, 20133 Milano, Italy; laura.marinoni@crea.gov.it (L.M.); tiziana.cattaneo@crea.gov.it (T.M.P.C.)

**Keywords:** two-dimensional correlation analysis, capillary electrophoresis, proteomics, milk-whey

## Abstract

Proteomics and metabolomics are analytic tools used in combination with bioinformatics to study proteins and metabolites which contribute to describing complex biological systems. The growing interest in research concerning the resolution of these systems has stimulated the development of sophisticated procedures and new applications. This paper introduces the evolution of statistical techniques for the treatment of data, suggesting the possibility to successfully characterize the milk-whey syneresis process by applying two-dimensional correlation analysis (2DCOR) to a series of CE electropherograms referring to milk-whey samples collected during cheese manufacturing. Two cheese-making processes to produce hard cheese (Grana type) and fresh cheese (Crescenza) were taken as models. The applied chemometric tools were shown to be useful for the treatment of data acquired in a systematically perturbed chemical system as a function of time.

## 1. Introduction

Proteomics and metabolomics are two disciplines that contribute to describing biological systems. Proteomics defines patterns of proteins expressed in different biological samples at a specific moment. It allows quantitatively analyzing and identifying proteins whose expression differs according to the experimental conditions of interest. Proteomic applications can vary greatly from straightforward protein identification to the complex characterization of protein–protein interactions [1,2]. Metabolomics represents a comprehensive procedure for metabolites assessment that involves measuring the overall metabolites of biological samples [3,4].

Among liquid phase separation techniques, capillary electrophoresis (CE) allows the analysis of proteins and peptides of any size and solubility, many of which would not migrate or would not be retained or stained in ordinary polyacrylamide gels. Moreover, by CE also non-protein compounds (e.g., organic acids) and not-charged molecules can be analyzed, improving the information on the studied bio-system.

Proteomics is mainly applied in medical and biological fields, even if nowadays applications are reported in the literature also for food [5,6,7,8,9]. The information on the occurrence and amount of a particular protein or derived compounds is extremely useful in the assessment of processes and adulterations. Several papers have been published also in the dairy field [10,11,12,13,14,15,16,17,18,19,20]. Modifications such as heat denaturation or proteolysis, common in the manufacture of many dairy products, give rise to new complex compounds and smaller peptides and amino acids whose analysis is not easily performed.

In the last few decades, chemometrics had a major role in the progress of analytical chemistry. There is, however, a field in chemometrics, called two-dimensional correlation analysis (2DCOR) [21], which seems to be able to analyze the more advanced data sets and to create simple, easily understandable results.

This method was built to extend the possibilities of infrared spectroscopy (IR) with some of the two-dimensional concepts used in nuclear magnetic resonance spectroscopy (NMR). It was a great success and later proved to be useful with many probes other than IR. Despite the rapid growth of the field, 2DCOR operated mostly in spectroscopy, and only a few attempts have been made outside that. However, we strongly believe that its basic concept can be advantageous in chromatography as well.

This paper introduces the development of statistical techniques for the data treatment, suggesting the possibility to successfully characterize protein and acidic compounds during the milk-whey syneresis process. In particular, two-dimensional correlation analysis (2DCOR) [21] was applied coupled to CE analysis to discover the main quality indices and to explore the time-series of the syneresis process in different cheese productions. This approach could help in extracting hidden information as the whey syneresis process progresses. The formation and the hydrolysis trends of some proteic and acidic compounds could be identified and/or correlated: information is not available only after a simple analysis of electropherograms. Several algorithms have been built up for the application of 2DCOR, but Noda [22,23] is recognized as the father of this technique.

The syneresis process is extremely meaningful in cheese production since different kinds of cheese require keeping different amounts of whey inside the curd and the volume of whey expelled directly affects the specific sensory and functional properties of the final product [24,25,26,27].

Whey released from curd is made of an aqueous solution of (i) proteins, namely α-lactalbumin, β-lactoglobulin, serum albumin, immunoglobulins, (ii) enzymes, (iii) caseinomacropeptide (CMP, deriving from k-casein hydrolysis), (iv) proteose-peptones, in some cases (v) residual caseins, (vi) lactose, (vii) minerals, and (viii) organic acids. Its composition varies during syneresis according to changes that curd undergoes during coagulation and cheese making [27,28].

The aim of this work was to use 2DCOR for studying the complex cross-correlations among protein constituents occurring in whey during the syneresis process. Moreover, multivariate statistics allowed the study of the evolving protein and acidic patterns arising from the process.

There is a lack of literature about the application of 2DCOR to chromatographic techniques, such as CE, but it could be an interesting approach. The whey syneresis process was taken as an example of “perturbation”. A set of analytical signals collected under whey syneresis and converted by 2DCOR, can provide rich and useful information about the presence of coordinated or independent changes among signals, as well as relative directions and sequential order of signal intensity variations.

Grana type and Crescenza cheese productions were taken as a model of two different categories of cheese, hard and fresh cheese, respectively. Hard cheeses are made by cutting the curd into small grains and heating it, to facilitate the expulsion of whey. Conversely, fresh cheeses are made by cutting the curd into big cubes to retain the whey as much as possible.

## 2. Results and Discussion

Since data were acquired in a systematically perturbed chemical system, as a function of time, the application of cross-correlation methods to measured data may lock-in to the modulated signal and enhance the measurement information [21].

In Figure 1, the electropherograms of the first sampling point for each cheese type are shown.

As expected, the milk whey obtained from the Crescenza cheese-making process was mainly characterized by the presence of serum proteins. Conversely, the milk whey from Grana type process showed also little amounts of casein fractions.

2DCOR was used in connection with CE analysis to study Residual Protein Constituents (RPC) changes in the whey syneresis process during Crescenza and Grana Padano type cheese production. Protein components involved in the syneresis process during the production of these types of cheese appeared as auto-peaks along the diagonal of the 2D-contour map, but also as positive or negative cross-peaks (Figure 2).

This means that protein components are evolving in-phase, especially α-lactalbumin, β-lactoglobulin, and caseinomacropeptide (CMP). In fact, both α-lactalbumin and β-lactoglobulin concentrations decrease during the syneresis process as was highlighted by quantitative CE analysis (Table 1), so their cross-peaks are positive. The negative cross-peaks between them and CMP indicate that most likely the latter increases during the syneresis.

In the case of 2DCOR coupled with CE, only useful information inside the synchronous map was found, indicating that the proteic compounds varied progressively, as the process proceeds.

In this way, the time-dependent evolution and the subsequent relaxation of the analytical signals arising from various excited constituents of the system as a consequence of the applied perturbation may be observed.

The scale transposition for the Crescenza cheese-making proved the possibility to apply chemometrics independently from the amount of milk used for its production. In fact, Table 1 showed comparable trends and values for whey proteins during the syneresis process at both a lab and pilot scale.

This behavior reflected the influence of the applied technologies in cheese manufacture: time, temperature, acidity, cutting methodology, and pressure applied on curd affect the cheese water content, and any change of these parameters in cheese technologies can influence the composition of released whey.

Crescenza cheese is an example of fresh cheese. Its technology requires the curd cutting into quite big pieces, so the surface area through which syneresis can occur is limited. Only nearly 60% of whey is released before curd extraction and molding. The effect of pressure on syneresis is greatly seen at molding because the pressure due to gravity on the curd increases by a factor of about 30 [27].

Curd acidification contributes to pH lowering and expulsion of a great quantity of whey, too, but only nearly 2 h after syneresis started. Normally, the water-binding of casein varies between 2 and 6 g per gram of protein; an increase of acidity provokes demineralization of casein and lowers hydration of casein micelles. Consequently, it seems as if whey proteins released after renneting and shrinkage of curd were progressively diluted for a decrease in solvation or water binding of the material making up the gel, because of pH lowering [28]. Another study, made on the same samples [29], highlighted that by FT-NIR analysis applied during the syneresis process, the milk system undergoes rearrangements; especially the break and restoration of water-protein linkages that seemed to be the main phenomenon responsible for system stability.

Grana cheese technology is quite different from Crescenza’s one, and this influences the syneresis process. Unlike Crescenza, Grana cheese technology requires a more extensive cutting to obtain lower dimensions of curd grain and a higher pressure during curd mixing. In this case, the rate of whey loss increases rapidly during the first 20 min after cutting and more than 90% of whey is removed from the curd before deposition in molds.

During Grana cheese production drainage rate is highly enhanced also by curd cooking at 52 –55°C: the incubation temperature of the cut coagulum affects the syneresis rate which accelerates as curd incubation temperature increases. The high cooking temperature could be responsible, too, for α-lactalbumin and β-lactoglobulin reduction in whey, because of hydrophobic and covalent interactions established between heat-denatured whey proteins and casein fractions.

Normally, curd cooking also selects a thermophilic lactic acid starter culture with a good acidifying capacity which produces lactic acid and thus positively influences whey removal. Whey loss after curd extraction is anyway limited and can stop before the end of the acidification process [27,30,31].

In Figure 3, the obtained OA electropherograms of the first sampling point for each cheese type are shown. In the electropherograms, H_2_SO4 came from sample preparation before CE analysis, and it does not contribute directly to the cheese-making process.

The evolving protein (RPC) and metabolic (OA) patterns arising from the whey syneresis processes, and their possible correlations were highlighted by building the Hetero Analysis Correlation Maps, reported in Figure 4.

The 2D hetero correlation is a very intriguing possibility to compare two different types of analytical signals, obtained by using different analytical procedures for a system under the same perturbation. In this way, if there is any underlying commonality between the response patterns of individual system constituents monitored by two different procedures and under the same perturbation, it should be possible to detect the correlation even between the different classes of analytical signals.

It was evidenced that OA peaks appeared as negative ones, thus indicating that both in fresh and hard cheese detected soluble compounds, especially lactic acid, have shown an opposite behavior in comparison with protein components: protein content decreased as organic acids (OA) increased.

As reported previously, acid production during cheese manufacture combined with heating and stirring of the curd/whey mixture, causes the casein curd to shrink and expel moisture. The kind of acids involved in this process is a function of the starter cultures selected for each type of cheese [27].

The concentration of citric acid in whey does not change since only the flavor producers (*L. lactis* subsp. lactis and *Leuconostoc* spp.) in mesophilic cultures metabolize citrate and produce CO_2_ and diacetyl, which are important in determining the texture and flavor of some cheeses. Citrate is not used as an energy source, but it is metabolized rapidly by these starters only in the presence of a fermentable carbohydrate. Moreover, only some *Leuconostoc* species produce citrate from the fermentation of lactose. On the contrary, Lactic Acid Bacteria (LAB) starters require fermentable carbohydrate, which is lactose, for energy production and growth, and lactate is generally the main product of fermentation. As the coagulation process goes by, LAB grow and produce lactic acid which increases acidity, decreases pH, and enhances syneresis. This is especially true for Crescenza cheese which requires only *S. thermophilus* involvement during the acidification process, whose principal product is lactic acid.

As far as Grana-type cheese is concerned, a higher variety of organic acids and protein fractions are involved in the process. Grana-type cheese is produced using natural starters containing especially *L. helveticus*, *L. delbrueckii* ssp. lactis and *L. delbrueckii* ssp bulgaricus, *S. thermophilus* and *L. fermentum* in variable amounts [27,28], which could activate alternative metabolisms and produce different organic acids apart from lactic acid. Regarding RPC in whey, small quantities of casein could derive from the extensive cutting of the curd and remain in the first whey sample. Subsequently, during the shrinking of the curd, they could newly aggregate and thus disappear from the other whey samples [29,31].

## 3. Materials and Methods

### 3.1. Milk-Whey Samples

Crescenza cheese was made by 11 laboratory tests on 3 L of milk/each and 5 cheese-making processes at pilot scale level (150 L of milk/each), as described in a previous study [29]. Laboratory cheese making was made in 5 L beakers, while pilot-scale tests were performed in 200 kg vats. The scale-up was carried out to verify that the scale transposition had no influence on the milk coagulation process. After clotting, six points (Sn) were identified as critical points in the syneresis process and milk-whey samples were collected at fixed times: 5 min after the first curd cutting (S1, t = 5′), 10 min after the second curd cutting (S2, t = 15′), 30 min after shaping (S3, t = 45′), 60 min after the first turning (S4, t = 105′) and at different times during curd rest (S5, t = 165′ and S6, t = 225′). The sampling scheme is reported in Figure 5.

Grana-type cheese was produced only at the pilot scale since it is a semi-hard, long ripening, and large dimension cheese (35 kg). The pilot production was made in the typical copper vat filled with 500 L of milk, which led to two twins Grana cheese wheels. Experiments were replicated 12 times. Also, for the Grana cheese production, six sampling points (Sn) were identified to monitor the syneresis process, after clotting and cutting, as following reported: after 10 min from the first curd cutting (S1, t = 0), and every 10 min during curd cooking at 38 °C (S2, t = 10′), 43° (S3, t = 20′) and 48 °C (S4, t = 30′), at the end of curd cooking at 55 °C (S5, t = 40′) and at curd extraction (S6, t = 60′). The flowsheet is reported in Figure 6.

### 3.2. Capillary Electrophoresis (CE)

The CE analyses were performed with a Bio-Focus 3000 Capillary Electrophoresis System (Bio-Rad Laboratories, Richmond, CA, USA). Data were collected and processed using the Bio-Focus 3000 Integrator software. Peak areas (A), migration times (Tm), and normalized peak areas, expressed as A/Tm, were calculated. Individual pure standards of α-lactalbumin, β-lactoglobulin, citric and lactic acids (Sigma-Aldrich, St. Louis, MO, USA) were used for peak identification, and calibration curves were prepared for quantification. Standards of αs-casein (αs1 and αs0), β-casein (β-casein A1 and A2), and acetic acid (Sigma-Aldrich, St. Louis, MO, USA) were used for identification, but they were not quantified. CMP standard was not available, so it was identified based on bibliography data and on the analysis of hydrolyzed κ-casein solutions, but it was not possible to quantify it. All analyses were carried out in duplicate.

#### 3.2.1. Whey Proteins Analysis by CE

Residual protein constituents (RPC) were analyzed according to Recio and Olieman [32]. Whey samples (400 μL) were diluted 2:3 with sample buffer (pH = 8.6 ± 0.1) containing 167 mM Tris(hydroxymethyl)aminomethane (Sigma), 42 mM morpholino-propane-sulphonic acid (Sigma), 67 mM EDTA (Sigma), 17 mM dithiothreitol (Sigma), 6 M urea (Sigma) and 0.5 g/L methyl-hydroxy-ethyl cellulose. Samples were centrifuged at 10,000 rpm for 10 min. Sample injection was made by pressure (0.5 psi*20 sec). Separations were carried out at 38 °C on a coated capillary with a 50 cm effective length × 50 µm i.d. (Bio CAP XL Coated Capillary, Bio-Rad, Hercules, CA, USA). Twenty mM sodium citrate-0.32 M citric acid buffer (Sigma, pH = 3.0 ± 0.1) mixed 2:3 with 10 M urea was used as running buffer. Dithiothreitol was added to a final concentration of 1.6 mM. Electromigrations were run at a constant voltage of 25 kV for 75 min. The UV-visible detector was used at 214 nm.

#### 3.2.2. Organic Acids Analysis by CE

Organic Acids (OA) analysis was performed according to Monti, Ghiglietti, and Cattaneo [33]. Twenty-five mL of 9 mM H_2_SO_4_ were added to 1 mL of milk-whey. The solution was filtered through a 0.20 µm disposable syringe membrane filter (Alltech, Milano, Italy), diluted 1:2 with 2 mM NaOH and degassed by centrifugation prior to CE analysis. Twenty mM Dipicolinic acid (pyridine-2,6-dicarboxylic acid) containing 0.5 mM Cetyltrimethylammonium bromide at pH 5.6 was used as a running buffer. Separations were carried out on a fused silica capillary (BioCAPTM Bare Silica Capillary, Bio-Rad, Hercules, CA) with 104 cm (100 cm effective length) × 50 µm. Samples were injected by pressure (5 psi*sec). Electromigrations were run at 20 °C with a constant voltage of −25 kV.

The signal wavelength was set at 230 nm and detection was performed in an indirect way. Bio-Focus 3000 Integrator software allowed organic acids peaks inversion, visualizing them as positive peaks and facilitating peaks integration.

### 3.3. Data Processing

#### 3.3.1. Electropherograms Alignment

To compare the behavior of milk-whey components during the syneresis from different cheese-making processes, the same CE peaks were aligned at the same elution time using the Grams/Ai ver. 6 software (Galactic Industries Co., Salem, NH, USA), and the obtained electropherograms were averaged according to the same kind of cheese-making (i.e., laboratory or pilot scale).

#### 3.3.2. Two-Dimensional Correlation Analysis (2DCOR)

Two-dimensional correlation analysis (2DCOR) was performed according to Noda [22] using Matlab ver. 7.14 software (The Math Works Inc., Natick, MA, USA) on the averaged electropherograms at each sampling point. The 2DCOR measures the correlations among variables that occur at the same frequency or rate and is typically used to characterize changes of a variable in connection with a dynamic parameter, in this case, the time. 2DCOR generates two kinds of spectra: synchronous and asynchronous. The synchronous spectrum contains information about features that change in phase, while the asynchronous spectrum displays information concerning out-of-phase covariance.

When 2DCOR is coupled with CE, the synchronous map shows constituents varying in the same direction (migration time): this is the map containing the most useful information. No information is in general observed inside the asynchronous map when biological processes are under study.

Peaks along the diagonal are named auto peaks: they are always positive and originate from the correlation of each variable with itself. Peaks that are off the diagonal, named cross-peaks, indicate the cross-correlation between two different variables and can be either positive or negative [21]. Positive cross-peaks result from features that are either increasing or decreasing together while the negative peaks indicate that one feature increases and the other decreases in intensity with respect to time.

#### 3.3.3. Hetero Analysis Correlation

The hetero analysis correlation is based on the same principle of 2DCOR and allows the comparison between analytical signals obtained by using different probes or procedures. It was applied in these experiments for calculations of correlation between whey proteins and organic acids analyses. In this context, the CE electropherograms of both RPC and OA were processed, as follow:the CE electropherograms related to the same sampling point were averaged;the organic acid profiles were used as inverted signals;Two matrices, one for OA and one for RPC, were built, where the migration times were used as columns, and the sampling points as rows; then the matrices were multiplied.

Thus, the main inputs for the correlation analysis were the migration times and the sampling points. Then, the generalized 2D correlation formalism developed and modified by Noda [23,34] was applied.

## 4. Conclusions

The application of two-dimensional correlation analysis (2DCOR) to CE electropherograms referring to proteins and organic acids variation during the whey syneresis process showed to be a good technique for the treatment of data acquired in a systematically perturbed chemical system as a function of time.

These results suggested the possibility to successfully add information for the description of complex systems and for the improvement of proteomic tools. The application of these techniques to other biochemical systems seems to be very promising.

Furthermore, the application of 2DCOR to different chemical patterns, inside the same biological system, could allow a rapid and clear collection of information about the presence of coordinated changes among analytical signals [34], as well as the relative directions and sequential order of the signal intensity variations.

## Figures and Tables

**Figure 1 molecules-26-04839-f001:**
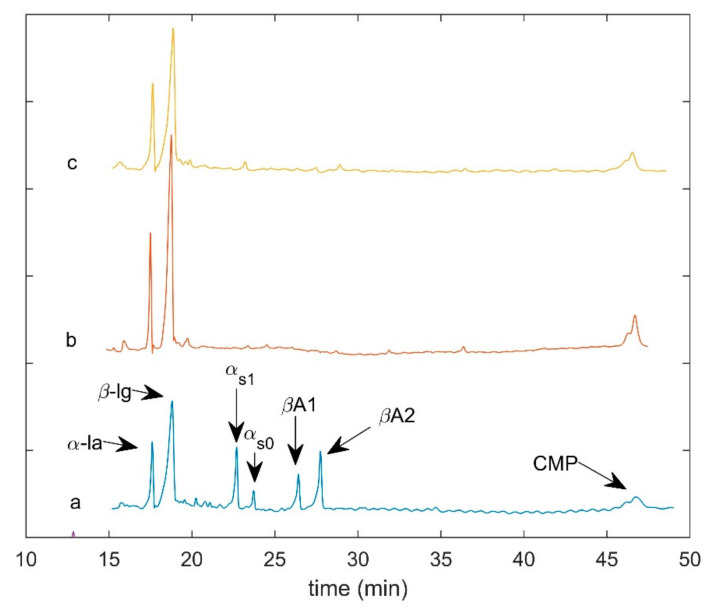
CE whey protein analysis: electropherograms of the first sampling point for each cheese type. (**a**) Grana type cheese; (**b**) Crescenza lab test; (**c**) Crescenza pilot scale. α-la (α lactalbumin); β-lg (β lactoglobulin); α_S1_ (α_S1_ casein); α_S0_ (α_S0_ casein); βA1 (β casein A1); βA2 (β casein A2); CMP (caseinomacropeptide).

**Figure 2 molecules-26-04839-f002:**
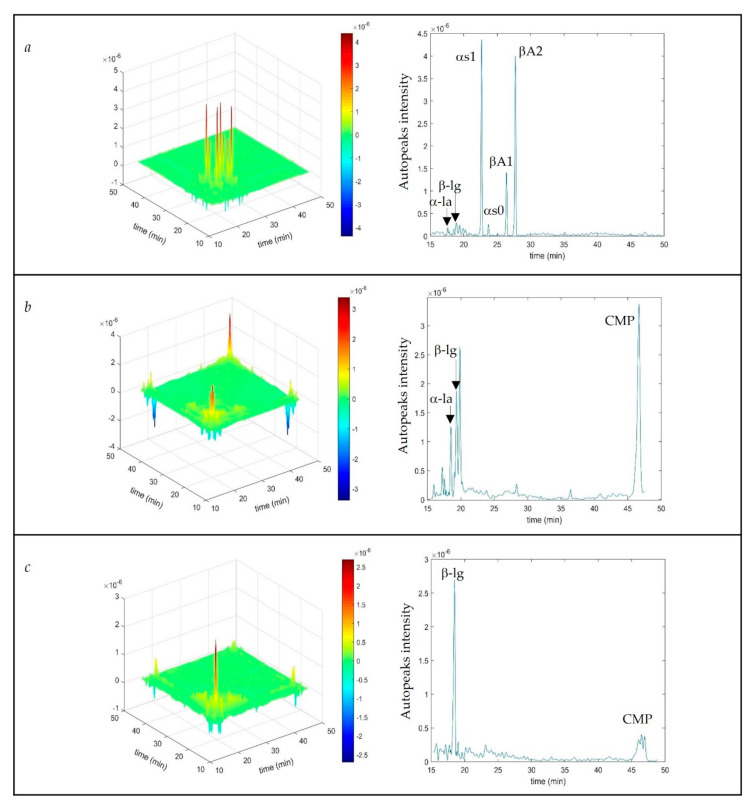
2DCOR (synchronous) of CE electropherograms of protein components for the studied cheese-making processes with migration times (left-hand) and the correspondent auto-peaks of the diagonal (CE migration time) of 2DCOR (right-hand). (**a**) Grana type cheese; (**b**) Crescenza lab test; (**c**) Crescenza pilot scale. Red color = high positive correlation.

**Figure 3 molecules-26-04839-f003:**
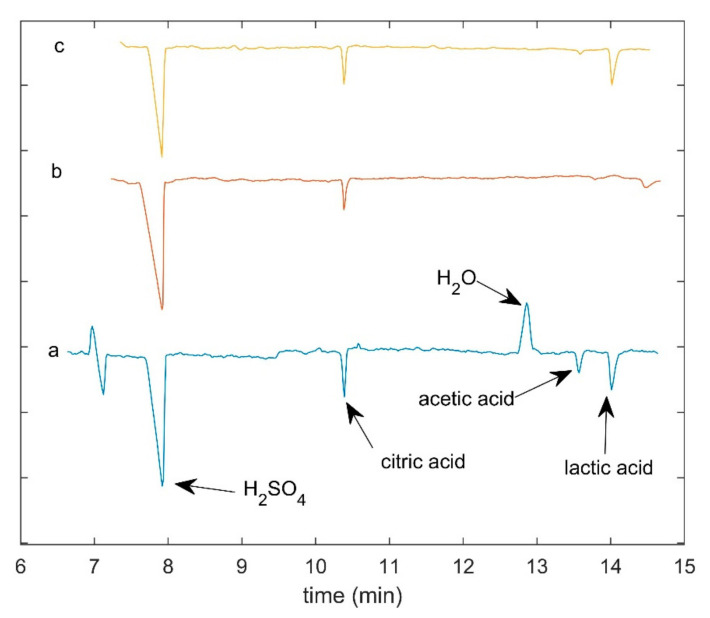
CE whey acids analysis: raw electropherograms of the first sampling point for each cheese type. (**a**) Grana type cheese; (**b**) Crescenza lab test; (**c**) Crescenza pilot scale.

**Figure 4 molecules-26-04839-f004:**
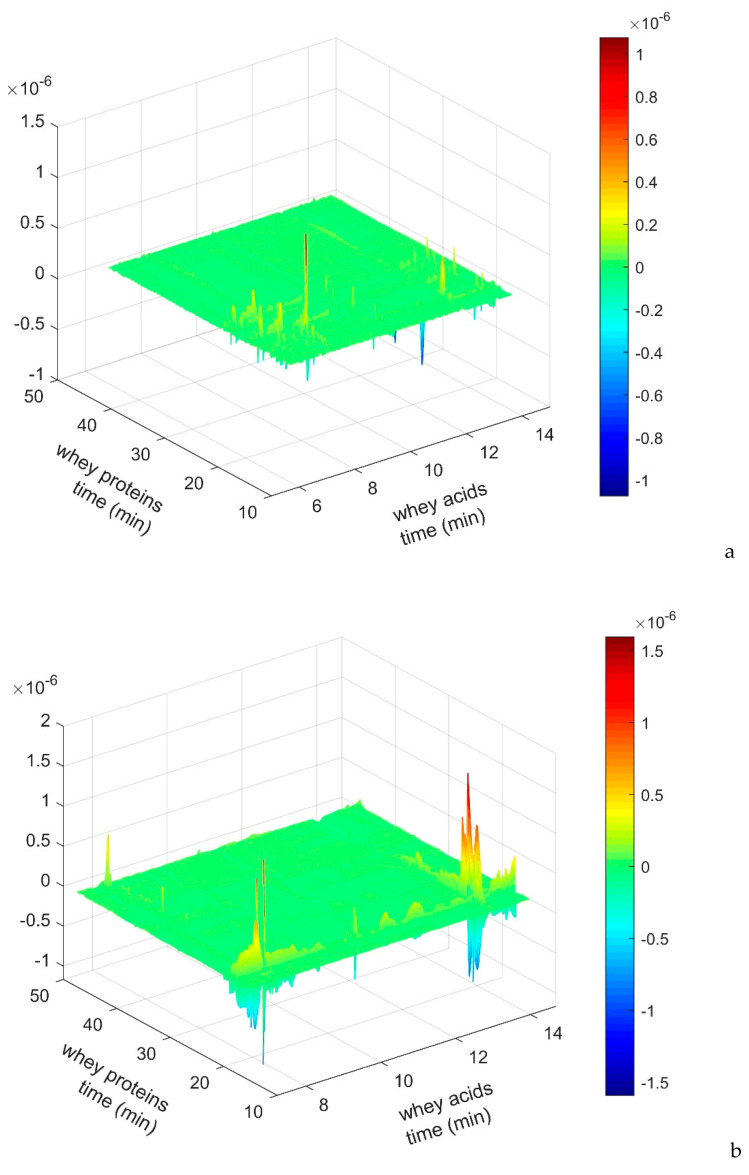
Hetero analysis correlation maps: RPC vs OA content. (**a**) Grana type cheese; (**b**) Crescenza lab test; (**c**) Crescenza pilot scale.

**Figure 5 molecules-26-04839-f005:**
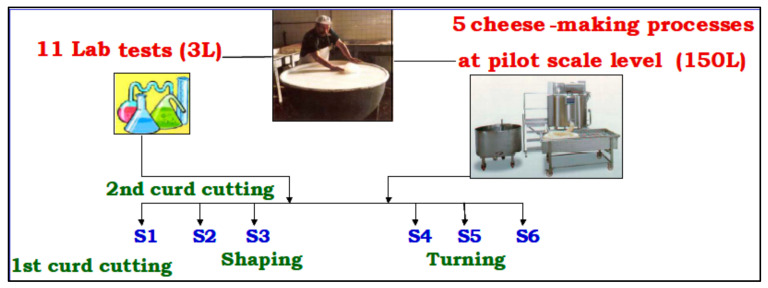
Crescenza syneresis process at the lab and pilot scale. S1–S6: sampling points.

**Figure 6 molecules-26-04839-f006:**
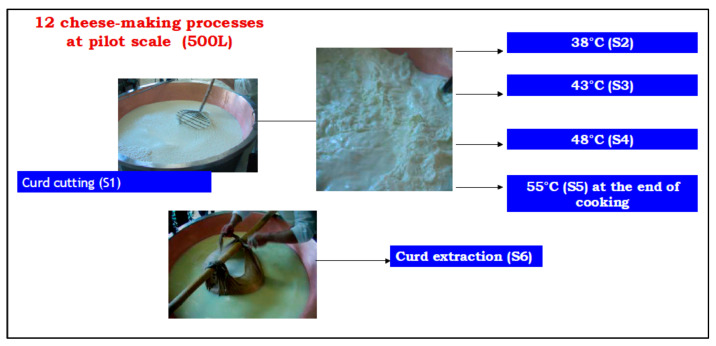
Grana-type cheese syneresis process at the pilot scale. S1–S6: sampling points.

**Table 1 molecules-26-04839-t001:** Milk-whey proteins and organic acids content during milk whey syneresis. S1–S6: sampling points (see Materials and Methods for details).

		α-lactalbumin g/L		β-lactoglobulin g/L		Citric Acid g/100g		Lactic Acid g/100g	
Crescenza laboratory tests	S1	1.93	±0.33	4.53	±0.95	0.12	±0.03	0.02	±0.00
	S2	1.70	±0.37	3.80	±0.87	0.12	±0.03	0.03	±0.00
	S3	1.46	±0.33	3.40	±0.87	0.11	±0.03	0.06	±0.01
	S4	1.31	±0.31	3.04	±0.83	0.10	±0.03	0.20	±0.06
	S5	1.10	±0.31	2.56	±0.77	0.11	±0.04	0.51	±0.02
	S6	1.12	±0.32	2.82	±0.82	0.13	±0.03	1.30	±0.17
Crescenza pilot scale level	S1	1.92	±0.19	3.87	±0.49	0.13	±0.04	0.03	±0.01
	S2	1.90	±0.32	3.89	±0.63	0.12	±0.04	0.05	±0.01
	S3	1.56	±0.12	3.20	±0.29	0.09	±0.05	0.19	±0.06
	S4	1.31	±0.19	2.69	±0.39	0.08	±0.04	0.40	±0.08
	S5	1.13	±0.32	2.37	±0.83	0.07	±0.04	0.66	±0.07
	S6	1.11	±0.33	2.23	±0.68	0.12	±0.07	0.92	±0.09
Grana Padano type	S1	1.78	±0.50	4.02	±0.92	0.15	±0.01	0.09	±0.01
	S2	1.63	±0.39	3.48	±0.80	0.13	±0.02	0.10	±0.01
	S3	1.60	±0.35	3.48	±0.64	0.14	±0.04	0.11	±0.02
	S4	1.36	±0.30	2.92	±0.70	0.14	±0.02	0.12	±0.01
	S5	1.36	±0.28	2.95	±0.70	0.18	±0.06	0.13	±0.01
	S6	1.27	±0.26	2.75	±0.46	0.14	±0.05	0.13	±0.01

## Data Availability

The study did not report any data.

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
