# Peer review of "Chemometrics for the Identification of Nitrogen and Acid Compounds in Milk-Whey as By-Products from Crescenza and Grana Padano Type Cheese-Making"

_molecules, 2021, doi:10.3390/molecules26164839_

Round 1
Reviewer 1 Report
Review – Manuscript Molecules 1308074, submitted to Molecules
The manuscript describes the investigation of the changes in the metabolome of different types of cheese during the cheese-making process using capillary electrophoresis and 2D-correlation techniques for data analysis. The paper is well written, the techniques used are sound and the conclusions are supported by the data. We have only minor suggestions to improve clarity and readability of the paper by a wider audience.
Small spelling and grammar corrections:
Line 58: “…suggesting the possibility to successfully characterizing...” change to “suggesting the possibility to successfully characterize”.
Lines 142-143: “and restoration of water-proteins linkages seemed to be one of the most responsible phenomena for system stability.”. Suggestion: “…seemed to be the main phenomena responsible for system stability.”
Specific Questions to be Adressed by Authors:
- Application of 2DCOR data analysis to capillary electrophoresis data is not commonplace since the technique is mainly applied to spectroscopy data. The authors give a brief overview of 2DCOR in the Introduction (lines 57-65, Introduction) but we suggest that authors discuss how 2DCOR can be applied specifically to CE and what kind of information can be extracted from it that cannot be easily obtained using just a simple analysis of electropherograms.
- In the “material and methods” section of the paper the authors describe that peak identification and quantification was made based on comparison with authentic standards. They list the following standards: α lactalbumin, β-lactoglobulin, citric and lactic acids. How the identification of the other metabolites (such as αS1 (αS1 casein); αS0 (αS0 casein); βA1(β casein A1); βA2 (β casein A2); and CMP (caseinomacropeptide) were made? Clarify in the text.
- In the analysis of Whey proteins (section 3.2.1) what was de injection method. Please specify in the text. Also, include sample size (volume of whey taken for dilution).
- The legend of Figure 2 is not very informative. Since application of 2DCOR to capillary electrophoresis data is scarce in the literature, we suggest that authors modify the legend to describe clearly what is represented in the left column (2DCOR analyses) and on the right column (autopeaks?). Can the peaks on the right column be associated with individual components based on migration times or they are just representing self-correlations? If possible, label peaks.
- Section 3.3.2, lines 282 and 283: “The synchronous spectrum contains information about features that change in phase, while the asynchronous spectrum displays information concerning out of phase covariance.”. This description applies to correlation of spectroscopy data where phase is a pertinent concept. What “in phase” and “out of phase” means regarding CE data. Does “in” or “out of phase” means two compounds varying in the same or opposite directions? Please clarify in the text.
- “Section 3.3.3 Hetero analysis correlation: For calculations of correlation between whey proteins and organic acids analyses the generalized 2D correlation formalism developed and modified by Noda [23] was applied.”. Please briefly describe the procedure to obtain the correlation results, in particular what data serves as input for the correlation analysis (migration time, area?).
Reviewer 2 Report
This study aimed at developing a nitrogen and acid profile in milk-whey for evaluating the syneresis process from Crescenza and Grana Padano type cheese productions. The research is interesting and maybe suitable for publication in this journal after crucial revision.
My comments are general, please see below: Very good title but the context and layout of Abstract and Introduction might be not relevant to the topics, especially regarding proteomics and metabolomics. Except for Abstract, Introduction and Conclusion, the Omics concepts are rarely mentioned in the whole manuscript. Instead, the research should emphasis on the CE coupled with 2DCOR to discover quality index of milk-whey and to explore the time-series of syneresis process in different cheese production. The Omics generally have the global changes of biological materials under a particular processing or treatment event. Actually it is a good manuscript without using Omics technology.
L30-31, post-translational modifications is a promising research target but it is too far away from this study.
L50-51, the literature No. 8 is not a Food-related research.
L57-62, L80-83, The sentence is too long and not clear.
L70-74, This paragraph should have cited references.
L126-128, Did the statistical analysis of data conducted in the Table 1? If possible, casein derivatives especially caseinomacropeptide (CMP), and acetic acid should be provided in the Table 1.
L162-166, Figure 3 showed a sharp peak for H2SO4 (E513?), and any contribution of the acid to syneresis, aggregation and coagulation in cheese-making?
L212-215, What is the differences between laboratory tests and pilot scale level besides scale-ups? What is the research motivation? Please elaborate more in Materials and Methods and/or Results and Discussion.
L216-232, Could the authors provide the approximate time periods (min / hr / days) at the S1-S6 stages for each scale and each cheese-making?
L223-232, Just curious about that it is not include the laboratory test for Grana cheese-making?
Please elaborate more in the motivation and discussion according to the most updated research.
